# Effect of Pre-Treatment with a Recombinant Chicken Interleukin-17A on Vaccine Induced Immunity against a Very Virulent Marek’s Disease Virus

**DOI:** 10.3390/v15081633

**Published:** 2023-07-27

**Authors:** Nitish Boodhoo, Ayumi Matsuyama-Kato, Sugandha Raj, Fatemeh Fazel, Myles St-Denis, Shayan Sharif

**Affiliations:** Department of Pathobiology, Ontario Veterinary College, University of Guelph, Guelph, ON N1G 2W1, Canada; boodhoon@uoguelph.ca (N.B.); matsuyam@uoguelph.ca (A.M.-K.); rajs@uoguelph.ca (S.R.); ffazel@uoguelph.ca (F.F.); stdenis@uoguelph.ca (M.S.-D.)

**Keywords:** MDV, chickens, Th17 cells, IL-17A, interferon-gamma and adaptive immunity

## Abstract

The host response to pathogenic microbes can lead to expression of interleukin (IL)-17, which has antimicrobial and anti-viral activity. However, relatively little is known about the basic biological role of chicken IL-17A against avian viruses, particularly against Marek’s disease virus (MDV). We demonstrate that, following MDV infection, upregulation of IL-17A mRNA and an increase in the frequency of IL-17A+ T cells in the spleen occur compared to control chickens. To elaborate on the role of chIL-17A in MD, the full-length chIL-17A coding sequence was cloned into a pCDNA3.1-V5/HIS TOPO plasmid. The effect of treatment with pcDNA:chIL-17A plasmid in combination with a vaccine (HVT) and very virulent(vv)MDV challenge or vvMDV infection was assessed. In combination with HVT vaccination, chickens that were inoculated with the pcDNA:chIL-17A plasmid had reduced tumor incidence compared to chickens that received the empty vector control or that were vaccinated only (66.6% in the HVT + empty vector group and 73.33% in HVT group versus 53.3% in the HVT + pcDNA:chIL-17A). Further analysis demonstrated that the chickens that received the HVT vaccine and/or plasmid expressing IL-17A had lower MDV-*Meq* transcripts in the spleen. In conclusion, chIL-17A can influence the immunity conferred by HVT vaccination against MDV infection in chickens.

## 1. Introduction

Since the discovery of murine and human T helper (Th) 17 cells, much progress has been made in defining their developmental origin, lineage, relative biological function, and respective expression of the IL-17 cytokine family members (IL-17A to IL-17F) [1,2]. It is clear that, in mice and humans, IL-17 has a role in immune responses associated with allergenic disease, autoimmune disease, malignancy, transplantation rejection, and modulation of host defense against both viral and bacterial microbes. While the expression of the avian IL-17 family of cytokines has been identified using intestinal intraepithelial lymphocytes (IELs), the functional role of these cytokines beyond differential gene expression has not been described [3].

IL-17 signaling is mediated through the IL-17A/F heterodimeric receptor complex (IL-17RA). The latter was demonstrated to have heterogeneous tissue distribution, which may serve to allow tissue-specific signal transduction [4]. As in mammalian hosts, in chickens, the IL-17RA molecule constitutes a distinct family of transmembrane proteins that are expressed by various immune system cells and fibroblasts [5,6]. IL-17A/IL-17RA interaction results in the induction of various cytokines ((IL-6, tumor necrosis factor (TNF)-α, and granulocyte monocyte colony stimulator factor (GM-CSF)), chemokines, antimicrobial peptides, and tissue-remodeling molecules. While Th17 cells are the best characterized source of IL-17, various innate immune system cells and T cell subsets can also express various IL-17 cytokines [7,8,9]. IL-17A expression may be preferentially induced in response to bacterial and fungal pathogens [4]. In fact, the functional role of IL-17A is much more understood in bacterial infections, while little or much less is known about its role in viral infection. In viral infection models, IL-17A overexpression modulates Th1/Th2 responses, leading to exacerbation of vaccinia virus virulence in infected mice [10]. In contrast, no direct anti-viral effects against herpes simplex virus 2 (HSV-2) were observed. However, its expression can stimulate CD4+ T cells to respond to HSV-2 reactivation in peripheral neurons [11,12]. Most of these studies were centered on non-avian hosts, and relatively little is known about the immune-protective function of chicken IL-17A against avian viruses [13].

Marek’s disease virus (MDV), also known as gallid herpesvirus 2 (GaHV-2), is the causative agent of Marek’s disease (MD) in chickens [14,15]. The clinical manifestation of MD is associated with transient paralysis, immunosuppression, metabolic dysregulation, and CD4+ T cell lymphoma formation in infected chickens [15,16,17]. Because of the cell-associated nature of MDV, T cell-mediated immunity is believed to be crucial in the control of MD [18]. The administration of chicken IFN-γ with MDV vaccine has been shown to increase vaccine efficacy, suggesting that IFN-γ may play an important role in boosting protection against MD [19]. IL-17 and IFN-γ play diverse roles and can modulate differentiation of the distinct Th1 and Th17 lineages, respectively [20]. That MDV infection modulates the host immune system suggests a potential underlying dueling role between IL-17 and IFN-γ in MD. The latter could indicate a relationship between pro-inflammatory cytokine expression and MDV infection [21,22]. Therefore, the aim of the present study was to identify and assess the potential role of IL-17A in MDV-infected chickens. To that end, cytokine expression from MDV-infected chickens, as well as tumor lesion severity, was assessed following treatment of chickens with recombinant IL-17A.

## 2. Materials and Methods

### 2.1. Plasmid and Cloning

The chicken IL-17A gene (accession number: NM_204460.2) was amplified from PMA/ION stimulated splenocytes by high-fidelity PCR (GoTaq polymerase; Promega, WI, USA) per the manufacturer’s recommendation and using designer primers (Table 1) to generate an amplicon (approximately 499 bp) with respective 5′-*Hin*dIII and 3′-*Eco*RV restriction sites. The gel purified amplicon (Gel purification Kit; Qiagen, Toronto, ON, Canada) was ligated into a pDRIVE vector (TA cloning; Qiagen, ON, Canada) and subcloned (*Hin*dIII and *Eco*RV restriction digestion; New England Bioscience, ON, Canada) into the pCDNA3.1/V5-HIS TOPO plasmid (Life Technologies, Mississauga, ON, Canada). DH5α competent cells were transformed and utilized to propagate the respective plasmids for purification with a midi-prep kit (Qiagen, ON, Canada). All plasmids (pCDNA3.1/V5-HIS TOPO and pCDNA3.1/rchIL-17A-V5-HIS TOPO) were linearized by *Bgl*II (New England Bioscience, Whitby, ON, Canada) digestion for in vitro transfection.

### 2.2. Cell Culture

*(i) Cells:* HEK 293-T cells (ATCC CRL-3216) were cultured and maintained (37  °C and 5% CO_2_) in minimum essential medium (MEM; Life Technologies, Mississauga, ON, Canada) supplemented with 10% fetal bovine serum (FBS; Gibco, Life Technologies, Canada) and 0.1% penicillin–streptomycin (Gibco, Life Technologies, Canada).

*(ii) Transfection/stably expressing cell lines:* One million HEK 293-T cells were seeded per well in a 6 well plate. Overnight cells (>80% confluent) were transfected, according to the manufacturer’s recommendation, using Lipofectamine stem reagent (Life Technologies, Canada) with up to 3 μg of linearized pCDNA3.1/V5-HIS-TOPO empty vector or pCDNA3.1/rchIL-17A-V5-HIS TOPO. After 48 h (hrs), cells were passaged (0.5% trypsin; Life Technologies, Canada) into T25 flasks and treated with 800 μg/mL G418 in MEM complete medium (10% FBS and 0.1% penicillin–streptomycin). The medium was changed every 48 hrs until proliferating islands of cells were observed. After 2–3 weeks of antibiotic selection, stably expressing cells were cultured without G418 for the purpose of collecting both cells and supernatant to confirm mRNA expression, as well as extracellular secretion of recombinant chicken IL-17A (rchIL-17A), and the cells were stored at −80 °C until required.

### 2.3. Virus Preparation

The very virulent MDV-RB1B strain (vvMDV), was provided courtesy of Dr. K.A. Schat (Cornell University, Ithaca, NY, USA) [17]. vvMDV-RB1B virus titers were calculated on primary chicken kidney cells obtained from 2- to 3-week-old specific pathogen free (SPF) chickens to establish infectious doses of inoculums, as well as stocks (liquid nitrogen storage) [23]. The HVT vaccine strain (MD-Vac-CFL; Fort Dodge Animal Health, Fort Dodge, IA, USA) was diluted in the recommended diluent and stored on ice prior to use.

### 2.4. Experimental Design and Sampling

*(i) Experimental animals:* Two hundred ninety-eight-day-old SPF chickens (layers) were purchased from the Animal Disease Research Institute, Canadian Food Inspection Agency (Ottawa, ON, Canada), and were accommodated in the isolation unit at the University of Guelph. For the duration of the experiments, all chickens were given ad libitum access to food and water. All experiments were approved by the Animal Care Committee of the University of Guelph and were conducted according to their guidelines.

*(ii) TLR ligands and MDV vaccine:* Synthetic class B CpG oligodeoxynucleotides 2007 (CPG ODN 2007; Invivogen, San Diego, CA, USA) and polyinosinic:polycytidylic acid (Poly I:C; Sigma-Aldrich, Oakville, ON, Canada) were re-suspended in phosphate-buffered saline (PBS, pH 7.4) and stored at −20 °C. The cell-free HVT vaccine (MD-Vac-CFL; Animal Health section, Boehringer-Ingelheim Canada, Burlington, ON, Canada) was resuspended in the recommended diluent and stored on ice until required for same-day inoculations.

*(iii) Experimental outline:* In the first experiment, 2-week-old chicks (n = 36) were injected intramuscularly (I/M) in the pectoral muscle with 100 μL of either CpG ODN 2007 (10 μg; n = 12) or Poly I:C (400 μg; n = 12). The control group (n = 12) received 100 μL of PBS (negative control). At 8 and 16 h post-inoculation, 6 chicks from each group were euthanized for sample collection.

In the second experiment, 1-day-old chicks (n = 108) were administered the HVT vaccine at one-half recommended dosage via a subcutaneous route. A reduced vaccine dose was used to observe the full potential of recombinant chicken IL-17A when used on its own or as a vaccine adjuvant. On day 3 post-vaccination, two groups of chicks were either inoculated intramuscular with 10 μg/chick of pCDNA3.1/V5-HIS-TOPO empty vector (n = 72) or pCDNA3.1/rchIL-17A-V5-His TOPO (n = 72). On day 4 post-vaccination, 198 chicks were infected with 250 plaque-forming units of the vvMDV-RB1B strain or sham/diluent treatment via the intraabdominal route.

*(iv) Sampling:* At various time points, tissue samples were collected for various analytical processes. Chicks (n = 6/group) from the first experiment were euthanized at 8 and 18 h post-TLR treatment, after which whole spleens were collected and stored (−20 °C) in RNA (Invitrogen, Toronto, ON, Canada). In the second experiment, whole spleens (n = 12/group) were collected in PBS containing penicillin (10 U/mL), and streptomycin (10 μg/mL) at 4, 10, and 21 days post infection (DPI), and parts of the whole spleen were either stored in RNA later for RNA extraction or were frozen in OCT embedding medium for immunostaining. Whole spleen samples were also collected from 5 control (non-infected chickens) and vvMDV-RB1B infected chickens at 4, 10, and 21 DPI and stored on ice for mononuclear cell isolation.

### 2.5. Spleen Mononuclear Cell Preparation and Stimulation

*(i) Mononuclear cells:* Whole spleens collected aseptically at various time points were applied onto 40-μm BD cell strainers (BD Biosciences, Mississauga, ON, Canada), crushed through using the rubber end of a 10-mL syringe plunger, and washed in RPMI 1640 with penicillin (10 U/mL) and streptomycin (10 μg/mL) [24]. Gradient suspensions were prepared by layering cells (2:1) onto histopaque 1077 (Millipore-Sigma, Oakville, ON, Canada) and centrifugation at 2100 rpm (600× *g* with no brakes) for 20 min to allow for the separation of mononuclear cells. Aspirated buffy coats were washed (2×) at 1500 rpm (400× *g*) for 5 min in RPMI 1640 with penicillin (10 U/mL), and streptomycin (10 μg/mL). Mononuclear cells were suspended in complete RPMI cell culture medium; RPMI 1640 medium contains 10% fetal bovine serum (Millipore-Sigma, Canada), penicillin (10 U/mL), and streptomycin (10 μg/mL). Cell numbers and viability were calculated using a hemocytometer and trypan the blue exclusion method. Mononuclear cells were suspended in complete RPMI cell culture medium at a density of 5 × 10^6^ cells/mL and kept on ice.

*(ii) In vitro stimulation assays:* Spleen mononuclear cells were seeded in 96-well U-bottom plates at a density of 1.0 × 10^6^ cells/200 μL and stimulated first with vehicle and then either CpG ODN 2007 (5 ug/mL) or lipopolysaccharides (LPS; 10 μg/mL). Cells were subsequently incubated (41 °C and 5% CO_2_) overnight (18 h), and supernatants were collected for anti-IL-17A enzyme-linked immunosorbent assay (ELISA). All stimulation assays were performed at least three times in triplicate on different days. Second, cells were co-incubated with supernatant from pCDNA3.1/V5-HIS-TOPO empty vector or pCDNA3.1/rchIL-17A-V5-HIS TOPO stably expressing cell lines on ice (15 min in the dark) for FACS analysis.

### 2.6. RNA Extraction and Reverse Transcription

RNA was extracted from in vitro mononuclear cell cultures, as well as spleens, using Trizol following the manufacturer’s instructions and treated with DNA-free DNase (ThermoFisher Scientific, Mississauga, ON, Canada) as previously described [25]. The quality, as well as the quantity, of RNA was estimated using NanoDrop ND-1000 spectrophotometry (NanoDrop Technologies, Wilmington, DE, USA). Subsequently, 1 μg of purified RNA was reverse transcribed to cDNA using Oligo(dT)12–18 primers (SuperScript II First-Strand Synthesis System; Invitrogen Life Technologies, Carlsbad, Ottawa, ON, Canada) according to the manufacturer’s recommended protocol. The resulting cDNA was diluted 1:9 in diethyl pyrocarbonate-treated (DEPC) water.

### 2.7. Real-Time Polymerase Chain Reaction (RT-PCR)

Quantitative real-time PCR with SYBR green was performed on diluted cDNA using a LightCycler 480 II (Roche Diagnostics GmbH, Mannheim, GER) according to the manufacturer’s recommendation. In brief, quantitative real-time PCR was performed with the following conditions: initial denaturation was performed at 95 °C for 5 min, followed by 40 cycles of denaturation at 95 °C for 10 s (s), primer annealing listed in Table 2 for 15 s, and extension at 72 °C for 20 s, with end-point melt-curve analysis. The relative fold change of target genes was calculated by 2−ΔΔCT method. The Ct value for each sample was normalized against the GAPDH housekeeping gene for respective samples. Data represent the means from six biological replicates, using primers outlined in Table 2.

### 2.8. Flow Cytometry

Following a wash in FACS staining buffer (PBS with 0.5% Bovine Serum Albumin; BSA), 5.0 × 10^5^ spleen mononuclear cells were counter-stained in two assays to demonstrate IL-17A functional interaction, as well as intracellular IL-17A cytokine staining. Antibodies used for the monocyte/macrophage panel (mouse anti-chicken Kul01-fluorescein isothiocyanate: FITC and mouse anti-chicken major histocompatibility complex (MHC) II-phycoerythrin: PE were all purchased from Southern Biotech, Birmingham, AL, USA) and T cell panel (mouse anti-chicken CD3ε-PB, mouse anti-chicken CD4-PE-CY7, mouse anti-chicken CD8α-FITC and mouse anti-chicken γδTCR-PE were all purchased from Southern Biotech, Birmingham, AL, USA).

*(i) rchIL-17A surface binding*: Mononuclear cells were incubated (4 °C for 15 min) in the presence of rchIL-17A. The cells were subsequently washed in FACS staining buffer (400× *g* for 5 min) and incubated (15 min at 4 °C) in the presence of unpurified mouse anti-chIL-17A antibody (IgG2a isotype), kindly provided by Dr. Thomas W. Göbel (Tierärztliche Fakultät, LMU München, Veterinärstrasse 13, 80539 München, Germany) [5]. The cells were washed again and incubated (15 min at 4 °C) in the presence of mouse anti-IgG2a-APC antibody prior to counter-staining per the APC or T cell panels listed. Following a final wash, mononuclear cells were counterstained with the live/dead marker 7-AAD (BDTM Pharmingen, Mississauga, ON, Canada) and acquired for analysis using a BD FACS Canto II.

*(ii) Intracellular cytokine staining:* IL-17 intracellular staining: Spleen mononuclear cells were washed in FACS staining buffer and subsequently counterstained (15 min at 4 °C) with the T cell panel antibodies. Following a wash, mononuclear cells were counterstained with the live/dead marker 7-AAD (BDTM Pharmingen, Canada). After washing twice with FACS staining buffer, the cells were incubated in 200 μL of Cytofix/Cytoperm solution (Beckton Dickinson Biosciences, Mississauga, ON, Canda) for 40 min at room temperature (RT), followed by washing three times with Perm/Wash solution (Beckton Dickinson Biosciences). The cells were stained with mouse anti-chIL-17A antibody (IgG2a isotype) for 15 min at RT in the dark and subsequently washed twice with Perm/Wash solution. Following final incubation (15 min at RT) in the presence of mouse anti-IgG2a-APC antibody, the cells washed in Perm/Wash solution and stored in FACS staining buffer.

IFN-γ intracellular staining: One million splenocytes were cultured with Golgi Plug and Dnase Ⅰ in a 96-well round-bottom plate for 4 h at 41 °C in 5% CO_2_. The cell staining protocol for flow cytometry was described elsewhere [32].

All samples were acquired on a BD FACS Canto II. Data were processed by FlowJo software, version V10.

### 2.9. Confocal Microscopy

*(i) Sample preparation:* OCT-embedded frozen spleen tissue samples were cut using a cryostat at approximately 10 µm in thickness. Samples were mounted onto slides and stored (−80 °C) for immunostaining. The slides were subsequently fixed (1 h in the dark at RT) in 4% paraformaldehyde (PFA). They were washed (PBS), permeabilized with 0.1% triton-X buffer solution (10 min in the dark at RT), and blocked (1 h in the dark at RT) with 1% bovine serum albumin (BSA-PBS). The slides were then washed 3× (PBS) and incubated (1 h in the dark at RT) with the mouse anti-chIL-17A antibody (IgG2a; 500 μL in 0.5% BSA–PBS). Following another wash (3× with PBS), the slides were again incubated overnight (4 °C in the dark at RT) with a goat anti-mouse IgG2a-488 (from ThermoFisher Scientific, ON, Canada) antibody (0.5 μg/500 μL in 0.5% BSA–PBS). The next day, the slides were washed (3× with PBS), and nuclei were labeled with Vectashield containing 4′,6-diamidino-2-phenylindole (DAPI from Vector laboratories/Cedarlane, ON, Canada). Tissue samples were sealed with round coverslips for fluorescence imaging.

*(ii) Visualization:* Tissue samples were viewed using a Leica SP5 laser scanning confocal microscope, and optical sections were recorded using either 405 or 488 nM with a numerical aperture of 1.4 or 1.25, respectively. All data were collected sequentially to minimize cross talk between fluorescence signals. The data are presented as maximum projections of each channel and overlaid for analysis with LAS AF software.

### 2.10. ELISA

Stably expressing rchIL-17A or vector control HEK 293-T cell culture supernatants, as well as diluent control-, CpG ODN 2007- and LPS-stimulated spleen mononuclear cell culture supernatants (100 µL), were incubated overnight at 4 °C with ELISA coating buffer. The next day, wells were washed (PBS) and blocked (5% BSA-PBS) for 1 h at RT prior to incubation (1 h at RT) with the mouse anti-chIL-17A antibody (IgG2a; 500 μL in 0.5% BSA–PBS). Following another wash (3× with 0.01% tween-PBS), the wells were again incubated (1 h at RT) with a goat anti-mouse IgG H/L-horse-radish peroxidase (HRP) antibody (0.5 μg/500 μL in 0.5% BSA–PBS from Southern Biotech, Al, USA). Following another wash (3× with 0.01% Tween-PBS), the wells were developed with the substrate (ABTS, peroxidase substrate system; Kirkegaard and Perry Laboratories, Gaithersburg, MD, USA). After 20−30 min of incubation, the optical densities were measured at 450 nm using an ELISA plate reader (Bio-Tek Instruments, Inc., Winooski, VT, USA).

### 2.11. Statistical Analysis

Graph-Pad Prism software, version 8 for Windows, was utilized to generate graphs and perform statistical analysis. All data are presented as means + standard deviations (SDs) and analyzed by Tukey’s post hoc test, the Kruskal–Wallis non-parametric test, Wilcoxon’s test (Mann–Whitney,) or Fisher’s exact test with the results shown as the mean ± standard deviation by Graph-Pad Prism software, version 8. The results were considered statistically significant at *p* < 0.05 (*).

## 3. Results

### 3.1. Expression of IL-17A, COX-2, TGF-β, and IFN-γ in vvMDV-RB1B Infected Chickens

The expression of IL-17A in the spleens of vvMDV-RB1B- and non-infected chickens was probed using confocal microscopy and real-time PCR (Figure 1). Our results demonstrated the detection of IL-17A in the spleens of vvMDV-RB1B-infected and not in non-infected chickens at 4 and 10 dpi (Figure 1A,B). IL-17A could be detected in the spleens of non-infected chickens at 21 dpi (Figure 1C). Based on punctate fluorescence, expression of IL-17A (488 nm) at the protein level was the highest at 21 days post infection (dpi) (Figure 1C) in vvMDV-RB1B-infected chicken when compared to 4 (Figure 1A) and 10 dpi (Figure 1B). The expression of IL-17A was also confirmed at the mRNA level based on real-time PCR (Figure 1D). The results demonstrated that IL-17A expression was significantly higher (*p* < 0.005) at 4 dpi compared to non-infected chickens. No difference was observed at 10 dpi. At 21 dpi, the expression of IL-17A was significantly lower (*p* < 0.005) compared to non-infected chickens.

The expression of COX-2 (Figure 1E), TGF-β (Figure 1F), and IFN-γ (Figure 1G) was also assessed. The results demonstrated upregulation of both COX-2 (Figure 1E) and TGF-β (Figure 1F) transcripts during the later stages of MDV-RB1B infection. COX-2 transcripts were significantly higher at 10 (*p* < 0.005) and 21 dpi (*p* < 0.005) compared to non-infected chickens (Figure 1E). Furthermore, TGF-β transcripts were significantly higher at 10 (*p* < 0.05) and 21 dpi (*p* < 0.005) compared to non-infected chickens (Figure 1F). During the early stages of MDV-RB1B infection, IFN-γ transcripts were increased (Figure 1G). At 4 dpi, IFN-γ gene expression was significantly higher (*p* < 0.005) in MDV-RB1B infected chickens but not at 10 or 21 dpi compared to non-infected chickens (Figure 1G).

### 3.2. Detection of IL-17A+ and IFN-γ+ T Cells Post vvMDV Infection

Spleen mononuclear cells were probed to define the IL-17A-producing cells at 4, 10, and 21 dpi (Figure 2). The gating strategy per FACS analysis defining intracellular IL-17A+ αβ or γδ T cells is shown (Figure 2A). Intracellular IL-17A cytokine staining was confirmed in spleen mononuclear cells of both control and vvMDV-RB1B-infected chickens at 4, 10, and 21 dpi (Figure 2B–D). The results demonstrated an increase in the frequency of CD3ε+CD8α+IL-17A+ γδ (Figure 2B), CD3ε+CD4+IL-17A+ (Figure 2C), and CD3ε+CD8α+IL-17A+ αβ T cells (Figure 2D). Specifically, the frequency of CD8α+IL-17A+ γδ T cells was significantly increased at 10 (*p* < 0.05) and 21 dpi (*p* < 0.01) compared to non-infected chickens (Figure 2B). αβ T cells were subdivided into CD3ε+ CD4+/CD8α+ αβ T cells. Within CD4+ αβ T cells, the frequency of CD3ε+CD4+IL-17A+ αβ T cells was significantly increased at 21 dpi (*p* < 0.05) and not at 4 or 10 dpi compared to non-infected chickens (Figure 2C). Within CD8α+ αβ T cells, the frequency of CD8α+IL-17A+ αβ T cells was significantly increased at 10 (*p* < 0.05), and 21 dpi (*p* < 0.01) compared to non-infected chickens (Figure 2D).

Splenocytes derived from vvMDV-RB1B- and non-infected chickens were cultured and IFN-γ production in various T cell subsets (αβ or γδ T cells) were analyzed by flow cytometry. We have previously shown, based on intracellular cytokine staining, that there is no significant difference in the frequency of CD8α+IFN-γ+ γδ T at 4, 10, and 21 dpi between vvMDV-RB1B- and non-infected chickens [32]. The gating strategy to define CD4+IFN-γ+ αβ T cells and CD8α+IFN-γ+ αβ T cells is shown (Figure 2A). The frequency of CD4+IFN-γ+ αβ T cells was significantly higher in vvMDV-RB1B infected chickens at 10 (*p* < 0.05) and 21 dpi (*p* < 0.01) than that in the control group (Figure 2E). IFN-γ production was significantly increased in CD8α+ αβ T cells derived from spleens of MDV-challenged chickens compared to control chickens at 4 (*p* < 0.05), 10 (*p* < 0.01), and 21 dpi (*p* < 0.05) (Figure 2F).

### 3.3. Cloning, Expression and Evaluation of the rchIL-17A Bioactivity Based on Receptor Binding

The full-length coding sequence of chIL-17A with signal peptide was subcloned into a pCDNA3.1/V5-HIS-TOPO plasmid by digestion/ligation reactions from a pDRIVE-IL-17A vector (TA cloning) (Figure 3A,B). HEK 293 T cells were transfected with the linearized pCDNA3.1/V5-HIS-TOPO and pCDNA3.1/rchIL-17A-V5-HIS TOPO to generate supernatant containing the recombinant chIL-17a from stably expressing cell lines. Expression of rchIL-17A in the culture medium was confirmed by an anti-V5 ELISA (Figure 3C). To demonstrate that a biologically active rchIL-17A protein (Figure 3B) is expressed by a eukaryotic expression system, chicken spleen mononuclear cells were co-incubated with supernatants from pCDNA3.1/V5-HIS-TOPO (control) or pCDNA3.1/rchIL-17A-V5-HIS-TOPO (IL-17A+)-transfected HEK-293 T cells. Spleen mononuclear cells were further phenotypically differentiated as either APCs (KUL01+MHCII+) or T cells (CD3+ αβ or γδ) per FACS analysis to demonstrate specific interactions (Figure 3D). Surface binding of rchIL-17A to its biological receptor, IL-17RA, was probed, and the gating strategy demonstrated detection of the rchIL-17A on the cell surfaces of both APCs and various T cell subsets following incubation with both control and IL-17A+ supernatant (Figure 3D). The results demonstrated that rchIL-17A can bind to a cell surface receptor found on both APCs and CD3+ T cells (αβ and γδ subsets). Co-incubation with the IL-17A+ supernatant delineated the frequency of APCs (Figure 3E), γδ (Figure 3F), and αβ (Figure 3G,H) CD3+ T cell subsets that express the cell surface IL-17RA molecule. Significantly higher (*p* < 0.05) frequencies of both CD4+ (Figure 3G) and CD8α+ (Figure 3H) αβ T cells were observed as expressing the IL-17RA complex.

### 3.4. Neither CpG ODN 2007 Nor Poly:IC Treatment Induce IL-17A Expression

The class B CpG ODN 2007 (5 μg/mL) and LPS (10 μg/mL) were used for in vitro stimulation of spleen mononuclear cells (Figure 4). After overnight culture, supernatants were collected for indirect ELISA and detection of IL-17A (Figure 4A). The results demonstrated that LPS, but not CpG ODN 2007, is an inducer of IL-17A expression. We further demonstrated that TLR3 or TLR21 signaling following intramuscular inoculation of 2-week-old chicks with either poly:(IC) or CpG ODN, respectively, did not lead to IL-17A expression at 8 and 18 hrs post-inoculation compared to PBS-treated chickens (Figure 4B).

### 3.5. Modulation of Cytokine Expression following rchIL-17A Inoculation

Three-week-old chicks were inoculated via the intramuscular route with 10 μg per chick of either the pCDNA3.1/V5-HIS-TOPO or pCDNA3.1/V5-HIS-TOPO-rchIL-17A plasmid (Figure 5). Twenty-four and 48 h later, whole spleens were collected for real-time PCR analysis. The differential gene expression analysis is presented for IL-1β (Figure 5A), TGF-β (Figure 5B), IL-2 (Figure 5C), IL-6 (Figure 5D), IL-17A (Figure 5E), IL-10 (Figure 5F), IL-12p40 (Figure 5G), and IFN-γ (Figure 5H). Pre-treatment with the rchIL-17A expressing plasmid modulated the expression of the various pro-inflammatory mediators compared to chicks treated with the control plasmid. The results demonstrated that expression of both IL-1β (Figure 5A) and TGF-β (Figure 5B) was significantly increased (*p* < 0.005) at 24, but not 48, hours post-treatment with the pCDNA3.1/rchIL-17A-V5-HIS TOPO plasmid compared to the pCDNA3.1/V5-HIS TOPO-treated chickens. No changes in IL-2 (Figure 5C), IL-6 (Figure 5D), and IL-17A (Figure 5E) mRNA transcripts were detected between pCDNA3.1/V5-HIS-TOPO-rchIL-17A- and pCDNA3.1/V5-HIS-TOPO-treated chickens. However, pre-treatment with the rchIL-17A overexpressing plasmid suppressed IL-10 (Figure 5F), IL-12p40 (Figure 5G), and IFN-γ (Figure 5H) gene expression. Specifically, the expression of IL-12p40 (Figure 5G) and IFN-γ (Figure 5H) was significantly decreased (*p* < 0.005) at both 24 and 48 hrs post-inoculation with the pCDNA3.1/rchIL-17A-V5-HIS TOPO plasmid compared to the pCDNA3.1/V5-HIS-TOPO-treated chickens. Expression of IL-10 (Figure 5F) was significantly decreased (*p* < 0.05) only at 48 h post-inoculation with the pCDNA3.1/rchIL-17A-V5-HIS TOPO plasmid compared to the pCDNA3.1/V5-HIS-TOPO-treated chickens.

MD Tumor Incidence in rchIL-17A-Treated and HVT-Vaccinated Chickens

Chicks were inoculated with 10 μg of the pCDNA3.1/V5-HIS-TOPO or pCDNA3.1/rchIL-17A-V5-HIS TOPO plasmids 24 hrs prior to infection with an MDV-RB1B virus, and they were necropsied at 21 dpi to assess the presence of MD lymphoma in visceral organs (Figure 6A). Tumor incidence defines the number of individual chickens that presented with tumors. One hundred percent of all chickens (15/15 chickens) infected with only the vvMDV-RB1B virus developed MD tumors, while no tumors were observed in the non-infected/non-vaccinated group (Figure 6B). The incidence of MD tumors was reduced in chickens that received half the dose of HVT + pCDNA3.1/V5-HIS-TOPO-rchIL-17A (50%; 7/15) compared to the groups of chicken that were vaccinated with half the dose of HVT and subsequently challenged with pCDNA3.1/V5-HIS-TOPO/RB1B (73.3%; 10/15 chickens) or RB1B only (66.6%; 11/15 chickens) (Figure 6B). The group of chickens that received only pCDNA3.1/V5-HIS-TOPO-rchIL-17A treatment before infection with vvRB1B had a higher tumor incidence (80%; 12/15) than chickens that received half the dose of HVT and were subsequently challenged with RB1B (73.3%; 10/15 chickens) (Figure 6B), although no significant differences were observed.

Further, at 21 dpi, the average lesion scores (the number of organs showing MD lesions per bird) within each group were evaluated (Figure 6C). The average lesion scores in chickens infected only with vvMDV-RB1B were significantly higher (*p* < 0.05) compared to chickens that were vaccinated (HVT) and subsequently infected (vvMDV-RB1B), vaccinated, (HVT)-IL-17A treated, and subsequently infected (vvMDV-RB1B) or vaccinated, (HVT)-pCDNA3.1 treated, and subsequently infected (vvMDV-RB1B). No differences were observed between chickens that were IL-17A treated and subsequently infected (vvMDV-RB1B) and those that were pCDNA3.1 treated and subsequently infected (vvMDV-RB1B) compared to vvMDV-RB1B infected chickens only (Figure 6C).

Replication of vvMDV-RB1B was assessed by real-time PCR for the expression of MDV-*Meq* in spleens at 21 dpi (Figure 6D). No MDV-*Meq* transcripts were detected in the control chickens. Chicks that were vaccinated (HVT) and infected (vvMDV-RB1B) had significantly lower (*p* < 0.05) MDV-*Meq* transcripts compared to chickens that were only vvMDV-RB1B-infected but not HVT+rchIL-17A+vvMDV-RB1B or HVT+pCDNA+vvMDV-RB1B-treated chickens at 21 dpi. No significant differences in MDV-*Meq* transcripts were observed between the HVT+vvMDV-RB1B-treated chickens and HVT+rchIL-17A+vvMDV-RB1B- or HVT+pCDNA+vvMDV-RB1B-treated chickens at 21 dpi. Furthermore, no significant differences in MDV-*Meq* transcripts were observed between the vvMDV-RB1B-infected chickens and rchIL-17A+vvMDV-RB1B or pCDNA+vvMDV-RB1B chickens.

Differential Cytokine Expression in Spleen of vvMDV-RB1B Infected Chickens Pre-Treated with the rchIL-17A

To demonstrate the potential modulatory effects of rchIL-17A in MD, real-time PCR analysis was used to assess the differential expression of genes (COX-2, IL-10, IL-17A, TGF-β, IL-12p40, and IFN-γ) in spleens at 21-dpi (Figure 7). The results demonstrated that chickens that were vaccinated (HVT) and/or treated with the rchIL-17A or empty vector and subsequently infected (vvMDV-RB1B) had significantly lower (*p* < 0.01) expression of COX-2 (Figure 7A), IL-10 (Figure 7B), and TGF-β (Figure 7C) compared to vvMDV-RB1B-infected chickens. COX-2 (Figure 7A) and TGF-β (Figure 7C; *p* < 0.01), but not IL-10 (Figure 7B; *p* < 0.05), transcripts were significantly lower in the rchIL-17A-vector-treated and subsequently infected (vvMDV-RB1B) group compared to the vvMDV-RB1B infected only chickens. Expression of IL-17A was significantly lower (*p* < 0.001) in the rchIL-17A- or empty-vector control and infected chickens (vvMDV-RB1B) compared to non-infected chickens (Figure 7D). HVT vaccination and/or pre-treatment with the rchIL-17A or empty vector followed by infection with vvMDV-RB1B resulted in an increase in IL-17A transcript levels (Figure 7D). No difference was observed between HVT and/or rchIL-17A- or empty vector-treated and infected chickens compared to the non-infected chickens (Figure 7D). IL-12p40 is essential for the induction of IFN-γ expression. While IL-12p40 expression was significantly increased compared to non-infected chickens (Figure 7E), no differences in IFN-γ transcripts were observed between vvMDV-RB1B-infected and non-infected chickens at 21 dpi (Figure 7F). Pre-treatment with the rchIL-17A or empty vector in vvMDV-RB1B-infected chickens did not result in changes in expression of either IL-12p40 (Figure 7E) or IFN-γ (Figure 7F) transcripts compared to vvMDV-RB1B-infected chickens only. The combination with HVT resulted in an increase in both IL-12p40 (Figure 7E) and IFN-γ (Figure 7F) transcripts. Specifically, chickens that were inoculated with HVT and/or that were rchIL-17A or empty vector treated and subsequently infected (vvMDV-RB1B) had significantly higher expression of both IL-12p40 (Figure 7E; *p* < 0.01) and IFN-γ (Figure 7F; *p* < 0.01) genes compared to the non-infected chickens.

## 4. Discussion

The IL-17A cytokine is a T cell-derived proinflammatory cytokine that has an important role in inflammation and immunity [1,7]. IL-17A is believed to be mainly produced by Th17 cells, which constitute a unique helper T-cell subset well defined in mice and humans [33]. While avian Th17 cells have yet to be characterized, IL-17A gene expression has recently been observed in mitogen stimulated avian αβ and γδ T cells and further detected via intracellular cytokine staining in various T cell subsets [5,6]. In this study, we demonstrated that: (i) infection with MDV resulted in early induction of IL-17A and IFN-γ expression in chicken splenocytes; (ii) based on FACS analysis, IL-17A+ T cells could be detected in splenocytes from vvMDV-RB1B infected chickens with the highest proportion of IL-17A+ T cells peaking at 21 dpi; and (iii) in vivo inoculation with a recombinant chIL-17A post-HVT vaccination significantly reduced tumor incidence compared to infected-only chickens compared to HVT+IL-17A or HVT+vector control. This outcome was associated with a reduction in COX-2, TGF-β, and MDV-*Meq* gene expression and an increase in IL-12p40 and IFN-γ expression.

The function and expression of IL-17A in the avian lung mucosal tissue following infection with MDV have not been previously defined. While the avian host’s early response (4 dpi) to MDV infection results in overexpression of both IL-17A and IFN-γ, their respective roles in chickens are not well understood. Due to the cell-associated nature of MDV, T cells play an essential role in immunity against this virus. We previously reported that the frequency of IFN-γ+CD8α+ γδ T cells in vvMDV-RB1B-infected chickens was significantly higher at 4 (*p* < 0.05) and 10 dpi (*p* < 0.01) than in the control group [32]. Pre-treatment with a recombinant chIFN-γ in an MDV infection model was associated with a reduction in tumor incidence [19]. However, subsequent IFN-γ knockdown based on an siRNA treatment had limited to no effects on MDV replication and tumor incidence [34]. More recently, Bertzbach et al. showed that chIFN-γ can have direct anti-viral effects against MDV and was induced by NK cells in vitro following MDV infection [35,36]. This finding indicated that other mechanisms exist that can limit MDV replication. In the present study, we demonstrated a relationship between the specific stages of MDV pathogenesis (4, 10, and 21 dpi) and percentage of IFN-γ+ and IL-17A+ T cells. Based on intracellular cytokine staining, the frequency of IFN-γ+ αβ T cells peaked at 10 dpi and was significantly lower at 4 and 21 dpi compared to 10 dpi. IL-17A+ T cells could be detected in splenocytes from vvMDV-RB1B infected chickens with the highest proportion of IL-17A+ αβ and γδ T cells peaking at 21 dpi. It is unclear whether the reduction in IFN-γ+ and increase in IL-17A+ T cells at 21 dpi relate to CD4+ T cell expansion and aggravation of lymphoma. In mice, IFN-γ is the major cytokine produced by CD27+ γδ T cells, and their CD27− counterparts preferentially secrete IL-17A [37,38]. In the present study, double positive IL-17A+IFN-γ+ T cells could not be defined due to limitations in antibody availability. Both IL-17A and IFN-γ can regulate the lineage differentiation of naïve T cells into Th1 or Th17 cells, respectively [33]. The presence of both IFN-γ+ and IL-17A+ T cells during MDV infection suggests a potential role in MDV pathogenesis.

The biological role of IL-17A has yet to be fully characterized in chickens but has been defined in mice and humans [39]. We observed a discrepancy between the abundance of IL-17A cognate protein and transcripts following MDV infection. Protein and RNA represent different steps of the multi-stepped cellular process, in which they are expressed, translated, and degraded. Specific regulatory processes can suppress mRNA transcription, thereby leading to no or limited protein expression. MDV is known to express and stimulate the expression of an array of micro RNAs that impact the host cellular machinery [40]. At mucosal sites, IL-17 may have protective roles by enhancing the Th1 response in host defenses against infectious diseases and promoting the induction of cytotoxic T lymphocyte (CTL) responses [33]. HVT-vaccinated chickens that were pre-treated with the rIL-17A had an increase in IL-12p40 and IFN-γ expression at 21-dpi compared to vvMDV-RB1B-infected chickens. This outcome was associated with a reduction in the expression of IL-10 and TGF-β at 21 dpi in IL-17A-pre-treated and HVT-vaccinated chickens. To gain some insight into the immune mechanisms triggered by treatment with IL-17A in chickens, we analyzed the cytokine expression profiles in the spleen. Both negative and positive feedback loops regulate the local and system effects of IL-17A.

Compared to negative control groups, vaccination with HVT does not result in a significant increase in IL-17A expression. Similarly, Hao et al. demonstrated that vaccination with the CVI988-RISPENS vaccine does not induce IL-17A expression [41]. This finding suggests that HVT vaccination elicits a cytokine-specific pattern of expression that works in synergy with pCDNA:IL-17A to modulate the IL-17A-mediated immunopathology observed post-MDV infection. In chickens, infection with a vvMDV-RB1B virus led to a significant increase in IL-17A expression at 4 dpi with no changes at 10 dpi and a significant reduction at 21 dpi compared to control chickens. It is possible that the increased production of proinflammatory cytokines in the absence of IL-17A during vaccination is a compensatory mechanism implemented to help protect the avian host against MDV infection. This theory suggests that IL-17A could be involved in modulating the lung mucosal environment. Viruses can utilize IL-17A to modulate the host immune system to support productive infection. *Gammaherpesviruses*, such as murine gammaherpesvirus 68 (MHV68), preferentially modulate the mouse immune system following primary infection, leading to IL-17A expression [42]. As such, during early infection, IL-17A can antagonize Th1 activity while promoting productive MHV68 viral infection. IL-17A directly suppresses the expression of IFN-γ, T-bet, and eomesodermin in T cells [43]. In MDV infection, we observed an increase in both IL-17A and IFN-γ during the early stage of infection. While antigen-specific T cell responses based on IFN-γ could be detected in MDV-infected chickens, it is not clear whether overexpression of IL-17A could limit antigen-specific T cell responses [18]. However, following MDV infection, IL-17A expression was induced during the early phase (4 dpi). More work is required to define whether IL-17A could be involved in the pathogenesis of MD.

In mammals, T cells of the γδ subset immediately produce IL-17A and induce inflammation upon pathogenic infections [44]. It is understood that immunity against herpes simplex virus type (HSV) 1 is partly imparted by the functional activity of IL-17A+ γδ T cells [45]. γδ T cells exert their immune-modulatory activity by secreting various innate factors, as well as Th1 and Th17 cytokines, which promote direct cytotoxic activity against infected cells [46]. During the early phase of MDV pathogenesis, high counts of both macrophages and T cells can be seen in the lungs of MDV-infected chickens [47]. Our FACS results indicated that γδ T cells are a major source of IL-17A during the later stages of MDV infection. It cannot be ruled out that the differentiation of IL-17A-producing γδ T cells occurs within the tumor microenvironment, but this possibility requires further experimental confirmation. IL-17A has been implicated in protective γδ T cell responses in other cancer models. Mechanistically, tumor-infiltrating IL-1β activated IL-17A+ γδ T cells, enhancing the priming and recruitment of IFN-γ-producing CD8+ T cells that exert antitumor effects [48]. We have previously reported that avian γδ T cells are activated during MDV infection and retain effector functions, based on expression of IFN-γ and cytotoxic (CD107a+) activity [32]. It is not known whether the increase in IL-17A at 4 dpi relates to an increase in T cell infection and modulation of antigen-presenting cell functions, such as macrophages [33]. However, IL-17 can promote the recruitment of innate cells to the site of inflammation [49]. We identified an increase in the percentage of IL-17A+ T cells at 21 dpi in the spleens of MDV-infected chickens. Chronic IL-17A production, at the protein level, has been shown to support tumor formation and progression [50]. This effect is derived from IL-17A-mediated signaling, which results in recruitment of innate immune cells to the site of injury [51]. IL-17-induced inflammatory mediators, such as IL-1β, IL-6, and TGF-β, promote a proangiogenic and immune suppressive tumor environment that enhances tumor growth. It is likely that the host response to MDV infection, which results in IL-17A progression, contributes to tumor formation. MDV is an oncogenic virus causing CD4+ T cell lymphoma in chickens. Transformation of CD4+ T cells has been linked to MDV-*Meq* function [52]. MDV-*Meq* has been shown to modulate the PI3K/AKT pathway [53]. In mice, the PI3K/AKT pathway regulates IL-17A expression and the induction of Th17 cells [54]. However, the mechanism for induction of IL-17A in MDV-infected chickens is unknown. Furthermore, it has yet to be shown whether IL-17A-mediated signaling during MDV results in tumor progression. Taken together, induction of IL-17A during MDV infection can have multifactorial effects.

## 5. Conclusions

In summary, our data revealed the effects of IL-17A during MDV infection and improved our understanding of adaptive immune responses against MDV. Preventive strategies that aim to induce strong Th1 cell immunity should be able to effectively control MDV infection. Along with Th1 responses, MDV infection also induces a Th17 immune response. More work is required to demonstrate this response in avian Th17 cells and the modulatory activity of IL-17A in avian innate and adaptive immune cell types.

## Figures and Tables

**Figure 1 viruses-15-01633-f001:**
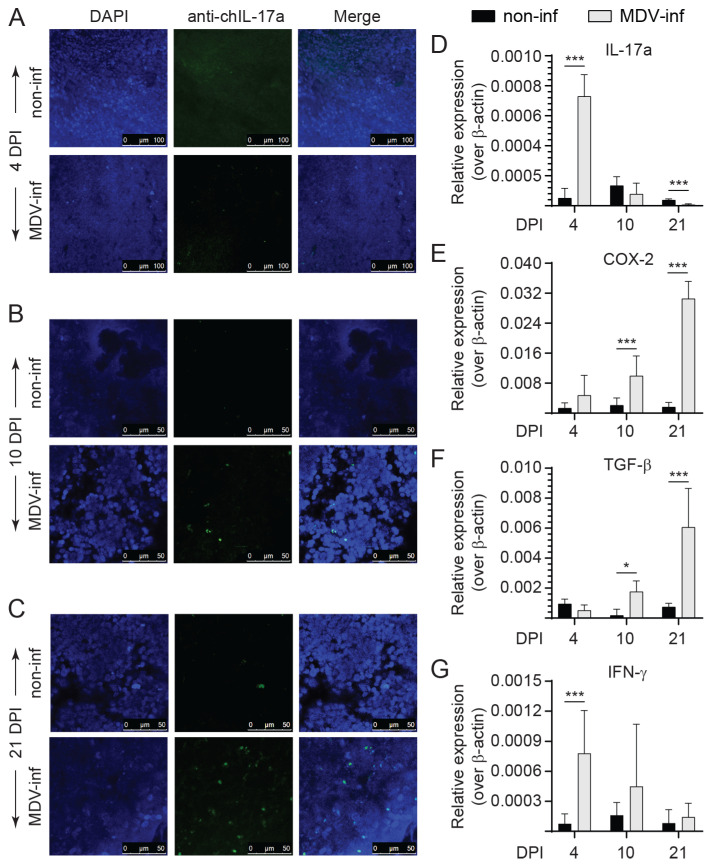
Expression of IL-17A, COX-2, TGF-β, and IFN-γ in vvMDV-RB1B-infected chickens. Representative pictures from confocal microscopy imaging with maximum projections of z-stacks for each channel demonstrating the localization of IL-17A (anti-IL-17A antibody; 488nm) in vvMDV-RB1B- and non-infected chickens at (**A**) 4, (**B**) 10, and (**C**) 21 dpi. Images are presented as maximum projections of all combined channels (merge). Real-time PCR analysis of (**D**) IL-17A, (**E**) COX-2, (**F**) TGF-β, and (**G**) IFN-γ gene expression at 4, 10, and 21 dpi in whole spleens. Target and reference gene expression was quantified by real-time-PCR, and expression is presented relative to β-actin expression. The results are based on at least 5–6 biological replicates in each group. Wilcoxon’s non-parametric test (Mann–Whitney) was used to test significance with the results shown as means ± standard deviations. * (*p* ≤ 0.05) and *** (*p* ≤ 0.005) indicates a significant difference.

**Figure 2 viruses-15-01633-f002:**
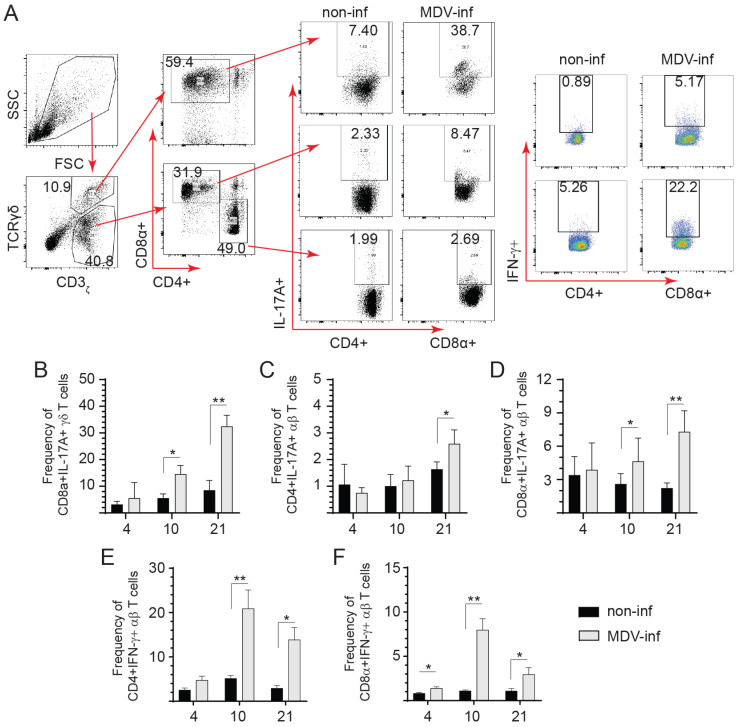
Frequency of IL-17A+ and IFN-γ+ T cells in spleens of vvMDV-RB1B-infected chickens at 4, 10, and 21, dpi. Representative FACS plots showing the frequency of IL-17A+ and IFN-γ+ T cells in the spleens of vvMDV-RB1B-infected chickens compared to non-infected chickens. (**A**) Dot plots demonstrate the gating strategy for identification of various CD3ε+ T cell subsets from vvMDV-RB1B- and non-infected chickens at 4, 10, and 21 dpi. The percentage of intracellular IL-17A+ (**B**) CD3ε+CD8α+ γδ T cells, (**C**) CD3ε+CD4+, and (**D**) CD3ε+CD8α+ αβ T cells in spleens from vvMDV-RB1B- and non-infected chickens are shown. Percentages of (**E**) CD4+IFN-γ+ αβ T cells within CD4+ αβ T cells and (**F**) CD8α+IFN-γ+ αβ T cells within CD8α^+^ αβ T cells. Wilcoxon’s non-parametric test (Mann–Whitney) was used to test significance. Data represent means ± standard deviations (n = 5) at each time point. * (*p* < 0.05) and ** (*p* < 0.01) indicates a significant difference.

**Figure 3 viruses-15-01633-f003:**
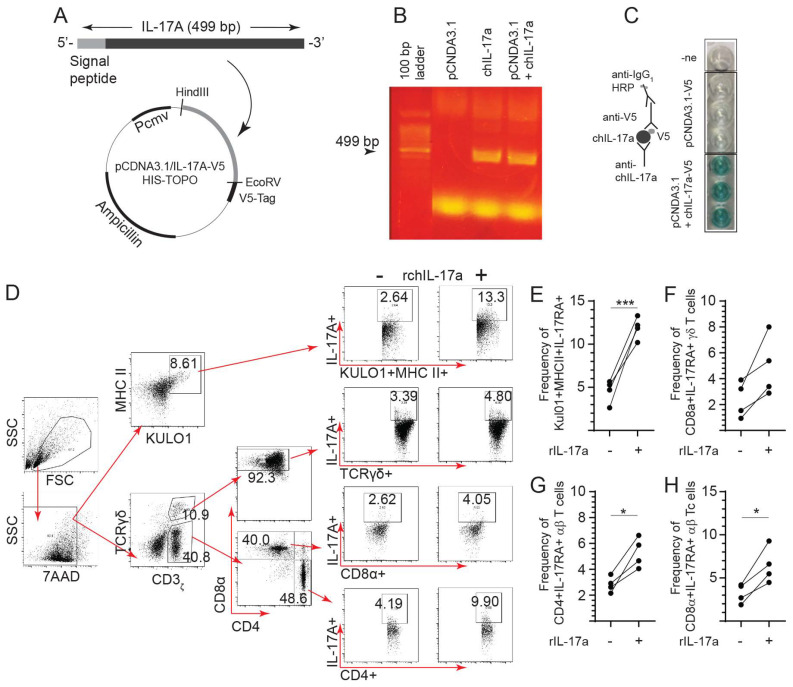
Overexpression of recombinant chicken IL-17A (rchIL-17A). Cloning of complete coding sequence of chicken IL-17A from PMA/ION-stimulated splenocytes into a (**A**) pCDNA3.1/V5-HIS-TOPO plasmid at the HindIII and EcoRV ligation sites. HEK-293 T cell were either mock-transfected (with empty vector) or with the recombinant plasmid (pCDNA3.1/V5-HIS-TOPO-rchIL-17A) to generate cell lines stably expressing IL-17A. Both cells and culture supernatant were collected 24 and 72 hrs post-transfection, respectively, to confirm expression of rchIL-17A at both (**B**) RNA level and (**C**) protein level. (**B**) Data shown are representative gel electrophoresis for visualization of rchIL-17A amplicon following standard PCR. (**C**) Indirect ELISA demonstrating extracellular expression of rchIL-17A. (**D**) FACS dot plot showing gating strategy to confirm bioactivity by functional interaction with the IL-17RA on (**E**) KUL01+MHCII+ (antigen presenting cells), (**F**) CD3ε+CD8α+ γδ T cell, and (**G**) CD3ε+CD4+ and (**H**) CD3ε+CD8α+ αβ T cells in live mononuclear cells by 7-AAD exclusion. The results are based on at least 5–6 biological replicates in each group. Wilcoxon’s non-parametric test (Mann–Whitney) was used to test significance with the results shown as means ± standard deviations. * (*p* ≤ 0.05) and *** (*p* ≤ 0.001) indicates a significant difference.

**Figure 4 viruses-15-01633-f004:**
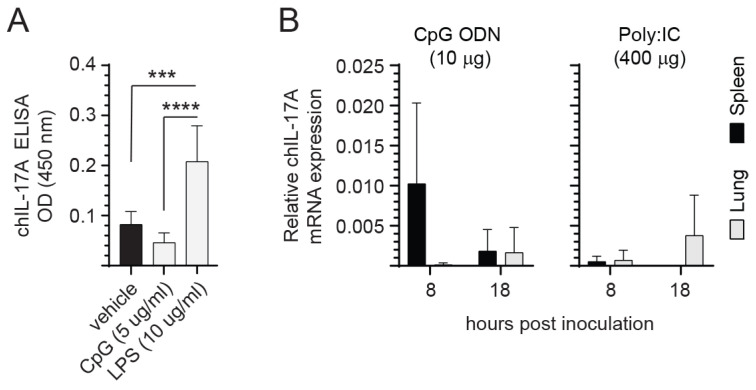
TLR ligands and IL-17a cytokine expression. The effect of TLR ligands on IL-17A expression both in vitro and in vivo. (**A**) Spleen mononuclear cells were isolated from 10-day-old chicks (n = 6) for in vitro stimulation with the TLR ligands CpG ODN 2007 and LPS. (**B**) Real-time PCR analysis for IL-17A expression in 2-week-old chicks (n = 36) injected intramuscularly (I/M) with 100 μL of either CpG ODN 2007 (10 μg) or Poly I:C (400 μg), respectively. Target and reference gene expression was quantified by real-time-PCR, and expression is presented relative to β-actin expression. The results are based on 12 biological replicates in each group. Wilcoxon’s non-parametric test (Mann–Whitney) was used to test significance with the results shown as means ± standard deviations. *** (*p* ≤ 0.001) and **** (*p* ≤ 0.0001) indicates a significant difference.

**Figure 5 viruses-15-01633-f005:**
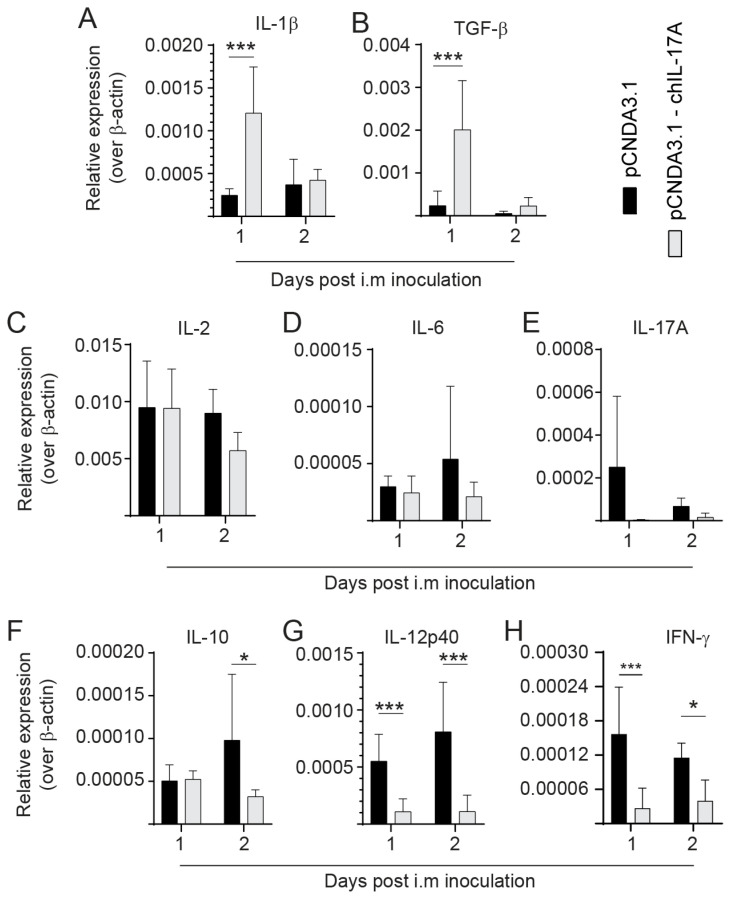
Differential gene expression in chickens pre-treated with the rchIL-17A overexpressing plasmid. Three-week-old chicks (n = 24) were inoculated via intramuscular route with 10 μg per chick of either the pCDNA3.1/V5-HIS-TOPO or pCDNA3.1/V5-HIS-TOPO-rchIL-17A plasmid (Figure 5). Twenty-four (n = 12) and 48 h later (n = 12), whole spleens were collected for real-time PCR analysis. The differential gene expression analysis is presented for (**A**) IL-1-β, (**B**) TGF-β, (**C**) IL-2, (**D**) IL-6, (**E**) IL-17A, (**F**) IL-10, (**G**) IL-12p40, and (**H**) IFN-γ. Target and reference gene expression was quantified by real-time PCR, and expression is presented relative to β-actin. The results are based on 6 biological replicates in each group. Wilcoxon’s non-parametric test (Mann–Whitney) was used to test significance with the results shown as means ± standard deviations. * (*p* ≤ 0.05) and *** (*p* ≤ 0.001) indicates a significant difference.

**Figure 6 viruses-15-01633-f006:**
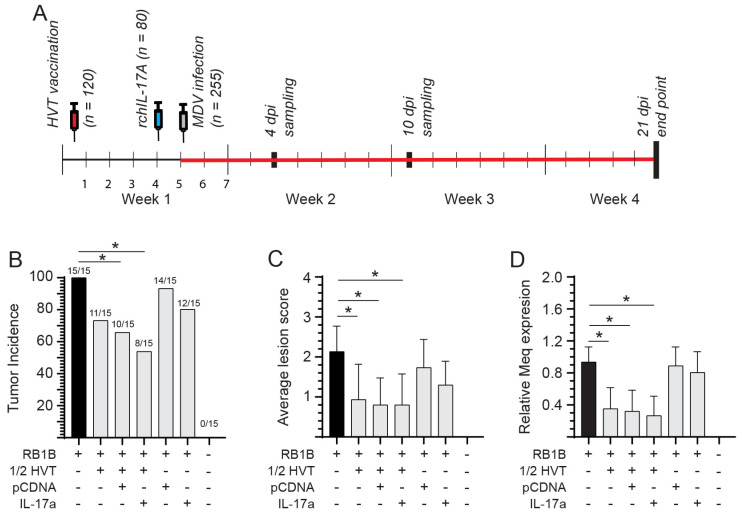
MDV tumor incidence at 21 dpi. Assessment of MD disease severity in vvMDV-RB1B-infected chickens based on presence of gross tumor lesions. (**A**) Schematic diagram depicting the experimental timeline for various treatments within each group. Day-old chicks were randomly allocated to each group. Sampling was performed at 4, 10, and 21 dpi. (**B**) Tumor incidence in the various groups was calculated by observing gross tumors in visceral organs of birds at 21 dpi. (**C**) Assessment of lesion score in different treatment groups based on the number of visceral and thoracic organs showing MDV-tumor lesions were counted in each bird, and the average score was calculated. (**D**) vvMDV-*Meq* gene expression at 21 dpi as quantified by real-time PCR and presented relative to β-actin expression. The results are based on at least 15 biological replicates in each group. (**B**) Fisher’s exact test or (**C**,**D**) Wilcoxon’s non-parametric test (Mann–Whitney) was used to test significance with the results shown as means ± standard deviations. * (*p* ≤ 0.05) indicates a significant difference.

**Figure 7 viruses-15-01633-f007:**
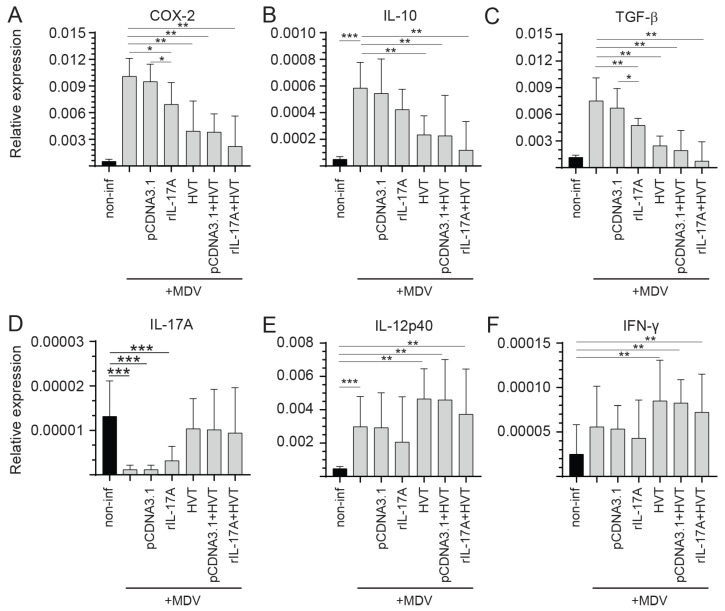
Expression of various cytokines in vvMDV-RB1B-infected chickens pre-treated with the rchIL-17A overexpressing plasmid at 21 dpi. Gene expression analysis at 21 dpi in vvMDV-RB1B-infected chickens that were pre-treated with the rchIL-17A overexpressing plasmid. Expression of the target genes (**A**) COX-2, (**B**) IL-10, (**C**) IL-17A, (**D**) TGF-β, (**E**) IL-12p40, and (**F**) IFN-γ was quantified by real-time PCR, and expression is presented relative to the reference gene (β-actin) in whole spleens. The results are based on at least 6 biological replicates in each group. Wilcoxon’s non-parametric test (Mann–Whitney) was used to test significance with the results shown as the means ± standard deviations. * (*p* ≤ 0.05) ** (*p* ≤ 0.01) and *** (*p* ≤ 0.001) indicates a significant difference.

**Table 1 viruses-15-01633-t001:** Primer sequences used for high-fidelity PCR.

Target	Primer Sequences		Accession No.
Full length chIL-17A	FWD	**AAGCTT**ATGTCTCCGATCCCTTG	NM_204460.2
	REV	**GATATC**AGCCTGGTGCTGGATCAGTGGG	

**Table 2 viruses-15-01633-t002:** Primer sequences used for real-time PCR.

Target	Primer Sequences		Reference
IL-1β	FWD	GTGAGGCTCAACATTGCGCTGTA	[23]
	REV	TGTCCAGGCGGTAGAAGATGAAG	
TGF-β	FWD	CGGCCGACGATGAGTGGCTC	[26]
	REV	CGGGGCCCATCTCACAGGGA	
COX-2	FWD	CTGCTCCCTCCCATGTCAGA	[16]
	REV	CACGTGAAGAATTCCGGTGTT	
IL-2	FWD	TGCAGTGTTACCTGGGAGAAGTGGT	[27]
	REV	ACTTCCGGTGTGATTTAGACCCGT	
IL-6	FWD	CGTGTGCGAGAACAGCATGGAGA	[28]
	REV	TCAGGCATTTCTCCTCGTCGAAGC	
IL-10	FWD	TTTGGCTGCCAGTCTGTGTC	[29]
	REV	CTCATCCATCTTCTCGAACGTC	
IL-12p40	FWD	CCAAGACCTGGAGCACACCGAAG	[28]
	REV	CGATCCCTGGCCTGCACAGAGA	
IL-17A	FWD	TATCAGCAAACGCTCACTGG	[30]
	REV	AGTTCACGCACCTGGAATG	
IFN-γ	FWD	ACACTGACAAGTCAAAGCCGCACA	[26]
	REV	AGTCGTTCATCGGGAGCTTGGC	
MDV-*Meq*	FWD	GTCCCCCCTCGATCTTTCTC	[23]
	REV	CGTCTGCTTCCTGCGTCTTC	
β-actin	FWD	CAACACAGTGCTGTCTGGTGGTA	[31]
	REV	ATCGTACTCCTGCTTGCTGATCC	

## Data Availability

Not applicable.

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
