# Peer review of "Effect of Pre-Treatment with a Recombinant Chicken Interleukin-17A on Vaccine Induced Immunity against a Very Virulent Marek’s Disease Virus"

_viruses, 2023, doi:10.3390/v15081633_

Round 1

Reviewer 1 Report

The study demonstrates the relationship between IL17A expression and MDV infection and pathogenesis, which has not been described so far and brings novelty to this research area. The study investigates overall immune signature profile upon exogenous IL17A treatment of chickens prior to challenge with pathogenic MDV strain. It also looks at the immune responses upon HVT vaccination combined with IL17A treatment, which has high relevance to poultry industry and veterinary research sector. Overall the manuscript is well written and only minor changes to the text would be recommended.

Line 18: abbreviation missing (vv)

Line 23: plasmid expressing IL17A

Line 28-30: Authors could mention here that there are more than one IL17 (family of genes).

Line 91: Please add ATCC reference to HEK293T cells.

Line 108: Correct to "courtesy".

Line 276-78: This text should be removed.

Lines 359 and 363: Please change reference Figure 2 to Figure 3.

Lines 428-30: The authors could mention why they looked at IL17A expression.

Lines 463-65: Figure 6B shows tumor lesions scores, figure 6A shows tumor incidence. Please correct accordingly.

Lines 492-94: The way it's written suggests that Meq expression was significantly lower in HVT+MDV than in HVT + IL17A/pCDNA vector + MDV which is not the case looking at the data. Please correct accordingly.  

Line 510-12: The authors could mention in the discussion what this implies (self-inhibitory regulation loop?)

Line 547: Please change to "post vaccination"

Line 570: Please change "is" to "in".

Lines 581-83: Gene expression increase is also true for HVT only if you compare to MDV infected chickens only. Please rewrite to make the message clear.

Lines 583-85: Compared to what group? Please correct the text to make the message clear.

Lines 588-90: Please add a short comment on differences seen at the mRNA level (RT-qPCR) and protein level (confocal microscopy) at different time points, e.g. high IL17A mRNA levels at 4 dpi and no signal detected in bioimaging, and the other way round at 21 dpi with strong signal detected by microscopy and minimal levels of mRNA (Fig 1).

Lines 593-94: Could the author please make a comment on how they arrived at this conclusion?

Author Response

attached are the responses to the reviewer’s comments

Reviewer 2 Report

Boodhoo et al investigated if chicken IL-17A, an important component of the host response to pathogens, shows potential in protecting against MDV. They conclude that the overexpression of chIL-17A, in combination with an MDV vaccine, reduces tumor incidence and lowers meq transcripts in the spleens of infected chickens.

Major points:

In their title and abstract, the authors state that IL-17A influences vaccine-induced immunity against vvMDV. The point here is that they do not really provide evidence for this statement. In their second animal experiment, they show that there are significant differences in MDV-induced pathology between unvaccinated and vaccinated groups, but not between the HVT-only, the empty vector and the IL-17A groups (as shown in Figs 6 and 7). They should tone down their title, abstract and parts of the results/discussion/conclusions. Agreed?
Along those lines, the conclusion paragraph actually does not conclude anything other than “MDV infection also induces a Th17 immune response”, which has been shown in previous studies already. Next, the authors state that “more work is required to demonstrate avian Th17 cells and the modulatory activity of IL-17A on avian innate and adaptive immune cell types” and that “the mechanism for induction IL-17A expression in MDV-infection is unknown at this point”. Why do their data “therefore suggest that exogenous IL-17A supports an optimal Th1 cell response and overall protection against MDV infection”? They do not show conclusive data to support this claim.

References 15-19 are inappropriate self-citations that should be replaced by relevant literature on the background of MDV and clinical manifestation of the disease (references 15 – 18) and on the importance of T-cell-mediated immunity (reference 19).

Mounting evidence indicates that IL-17A plays a key role in the initiation and progression of tumorigenesis, metastasis, and viral infections. Moreover, it has been experimentally established that IL-17A curtails the maturation process of NK cells, thereby impeding their effectiveness in combating tumors and viral invasions. This (IL17-A’s tumor-promoting roles/NK cells in MDV infections) should be discussed in the manuscript…

The stats analyses should be double-checked. My understanding was that the Wilcoxon-Mann-Whitney test is specifically designed for comparing two groups. However, when the objective is to compare the locations of multiple groups using a similar statistical approach, the Kruskal-Wallis test is typically employed.

How do the authors explain their findings shown in Fig. 5E? I am surprised that the IL-17A levels are reduced in the pCDNA3.1-rchIL-17A group compared to the empty vector…

Lines 510f (and Fig. 7D): Why was the expression of IL-17A significantly lower in the rchIL-17A-group when compared to non-infected chickens (Figure 7D)? That makes me worry…

Lines 547f: The statement “In vivo inoculation with a recombinant chIL-17A prior to HVT vaccination significantly reduced tumor incidence” is not at all backed up with data. It would be a relevant reduction if the authors would have detected differences to the HVT-only control as well as the HVT + empty vector control (as stated above).+

Lines 580f: Where do the authors show that “IL-17A can have an antiviral effect against MDV”?

Lines 603: The authors write that “… following MDV infection, IL-17A expression is induced but not observed in vaccinated chickens”. Do they refer to the data in Fig. 7D? There, it looks like MDV infection reduces IL17-A levels while that reduction is absent in the vaccinated groups at 21 dpi.

Minor points:

Line 108: wrong reference style (Schat et al., 1982). The same applies to lines 112 and 155.

Lines 116f: More information on the breed/genetic background of the chickens should be provided.

Line 116f: The manuscript would benefit from a table or figure depicting the exact experimental outline of the second animal experiment incl. animal numbers and respective treatments.

Line 201: Remove an “a” in “Aalbumin”.

Lines 203f: It would be good to provide more detailed information on the antibodies used.

Lines 211f: It should be “Dr. Thomas W. Göbel (Tierärztliche Fakultät, LMU München, Veterinärstrasse 13, 80539 München)”.

Line 282f: Isn’t the green staining in the mock sample at 21 dpi a specific IL17-A staining?

Lines 284f: The discrepancy between IF (highest IL17-A levels at 21 dpi) and RT-qPCR (highest IL17-A levels at 4 dpi) should be explained.

In the legend to Fig. 3 (as well as 4, 5, and 7): update the information on the significance asterisks.

Fig. 3: What are the x- and y-axes of the eight blots in the middle (- rchIL-17a +)?

Fig. 6: indicate how many chicks were in the control groups (far right columns; 0/15, I assume from what is stated in the legend?

Line 558f: Include additional discussion referring to additional recent literature on the anti-MDV effects of IFN-γ.

The authors should be consistent with their use of US vs. British English. The same applies to word capitalization (e.g. “Vaccination” vs” vaccination”).

Minor editing of English language required.

Author Response

(The authors gave the same response as above.)

Round 2

Reviewer 2 Report

Although I am generally content with the modifications the authors have made, I do have additional comments that I believe should be taken into consideration:

I still see the “effect of overexpressing rchIL-17A” as an issue in the presented work (line 17 and elsewhere). How do the authors explain that the IL-17A levels are reduced in the pCDNA3.1-rchIL-17A group compared to the empty vector? How can they state then that the effects they see are due to an overexpression of rchIL-17A? That has to be conclusively discussed in the manuscript! The authors replied in their rebuttal letter that they found no (significant) differences in the IL-17A levels in chickens treated with the empty vector control compared to the IL-17A plasmid (Fig. 5E). How can they write of “overexpression” in vivo then (see lines 19, 137 for example)?

I am surprised by the numbers of animals that were used for this study. In Fig. 6A, the authors state that the group sizes were n=120, n=80 and n=225. Yet, only 12 chickens per condition were sampled at each sampling time point p.i. What happened to all the other chickens? Why were they vaccinated/treated/infected? That is a question that a critical reader may seek answers to.

Throughout the manuscript: meq (lowercase m, italics) is the gene and Meq (uppercase M, roman) the protein. See https://doi.org/10.3382/ps/pez095, for example. Please correct.

Line 279: For which data did the authors use the Kruskal-Wallis test? Please clarify.

Correct “Real-time PCR analysis of (D) IL-17A, € COX-2, …” (line 317f)

Correct “Percentages of €(E) CD4+IFN-γ+ αβ T cells within…” (line 361).

Correct “To demonstrate a biologically active rchIL-17A protein (Fig 3 and B) is expressed by…” (line 373f).

Correct “…APCs and various T cell subsets foll€wing incubation with…” (line 381).

Correct “…the cell s€rface IL-17RA… (line 286).

Add “although the differences were not statistically significant” or something similar to line 479f.

Correct “(Figure 6A) in line 480. That should be Figure 6B. The references to the specific panels of Fig. 6 have to be corrected throughout the paragraph.

Line 564: What do the authors mean by “ameliorated its performance”? Please revise.

Add reference(s) to lines 600-605.

Change “on” to “during” in “In summary, our data reveal the effects of IL-17A on MDV infection and improve our understanding of adaptive immune responses against MDV.” (line 659f).

Reference 1: Why do the authors cite an Erratum?

References 33 and 34 are the same!

Reference 36: Correct “Effects of Interferon-?? Knockdown on Vaccine-Induced Immunity against Marek???S Disease in Chickens”.

Previous criticisms that have not been addressed appropriately (please address these again):

Previous criticism: Mounting evidence indicates that IL-17A plays a key role in the initiation and progression of tumorigenesis, metastasis, and viral infections. Moreover, it has been experimentally established that IL-17A curtails the maturation process of NK cells, thereby impeding their effectiveness in combating tumors and viral invasions. This (IL17-A’s tumor-promoting roles/NK cells in MDV infections) should be discussed in the manuscript…
Author response: “as suggested, a sentence has been added We observed a discrepancy between the abundance of IL-17A cognate protein and transcripts following MDV infection. Protein and RNA represent different steps of the multi-stepped cellular process, in which they are expressed, translated and degraded. Specific regulatory processes can suppress mRNA transcription thereby leading to no or limited protein expression. MDV is known to express and stimulate the expression of an array of micro RNAs that impact the host cellular machinery.” On line 584-588.”
Clarification: The response does not address the criticism.

Previous criticism: Line 510f (and Fig. 7D): Why was the expression of IL-17A significantly lower in the rchIL-17A-group when compared to non-infected chickens (Figure 7D)? That makes me worry…
Author response: “In figure 7C, the control group was compared to the group that was pre-treated with chIL-17A and subsequently infected with MDV.”
Clarification: The previous criticism referred to Fig 7D, not 7C.

Previous criticism: The stats analyses should be double-checked. My understanding was that the Wilcoxon-Mann-Whitney test is specifically designed for comparing two groups. However, when the objective is to compare the locations of multiple groups using a similar statistical approach, the Kruskal-Wallis test is typically employed.
Author response: “The reviewer is right that for grouped analysis the Kruskal-wallis test is utilized. In this case, the significant symbols to not reflect a group analysis. Specific bars were compared to each other and for the purpose of simplifying the figure panel, the lines were merged to show that all the specific groups when compared to a particular group all have the same level of significance. The figures 6 and 7 has been remastered to show the specific statistical analysis/significance.”
Clarification: Panel 6B has not been updated.

See "Comments"

Author Response

response to reviewer's comments
